New records of immature aquatic Diptera from the Foulden Maar Fossil-Lagerstätte, New Zealand, and their biogeographic implications

http://orcid.org/0000-0003-1893-3215 Baranov Viktor O. 1 viktor.baranov@ebd.csic.es
http://orcid.org/0000-0001-8254-8472 Haug Joachim T. 2 3
Kaulfuss Uwe 4
1 Estación Biológica de Doñana, Consejo Superior de Investigaciones Científicas , Sevilla, Andalucia , Spain
2 Biocenter, Ludwig-Maximilians-Universität München , Munich, Bavaria , Germany
3 GeoBio-Center, Ludwig-Maximilians-Universität München , Munich, Bavaria , Germany
4 Department of Animal Evolution and Biodiversity, Georg-August Universität Göttingen , Göttingen, Lower Saxony , Germany
Żyła Dagmara
Electronic publication date: 2024 Feb 26
Publication date: 2024
Volume: 12
Electronic Location ID: e17014
Received 2023 Nov 21; Accepted 2024 Feb 5
Copyright: © 2024 Baranov et al.
Copyright year: 2024
Copyright holder: Baranov et al.
License: This is an open access article distributed under the terms of the Creative Commons Attribution License, which permits unrestricted use, distribution, reproduction and adaptation in any medium and for any purpose provided that it is properly attributed. For attribution, the original author(s), title, publication source (PeerJ) and either DOI or URL of the article must be cited.
License URL: https://creativecommons.org/licenses/by/4.0/

Keywords: Diptera, Pupae, Aquatic, Fossil record, Chironomidae, Chaoboridae, New Zealand

Funding: German Research Foundation DFG Grant 429296833 Volkswagen Foundation PeerJ Publication Fees through the CSIC Open Access Publication Support Initiative from the CSIC Unit of Information Resources for Research (URICI) Spanish State Agency for Innovation’s Ramon y Cajal fellowship RyC2021-032144-I This work was funded by the German Research Foundation (DFG grant 429296833). Joachim T. Haug received support from the Volkswagen Foundation (Lichtenberg Professorship). Viktor Baranov received funds supporting the payment of the PeerJ publication fees through the CSIC Open Access Publication Support Initiative from the CSIC Unit of Information Resources for Research (URICI). Viktor Baranov’s work is sponsored by the Spanish State Agency for Innovation’s Ramon y Cajal fellowship (RyC2021-032144-I), project title “Climate change in the past and present & Insect decline”. The funders had no role in study design, data collection and analysis, decision to publish, or preparation of the manuscript.

==============================
Background

The biogeographical and ecological history of true flies (Diptera) in New Zealand is little known due to a scarcity of fossil specimens. Here, we report a fauna of immature aquatic dipterans from freshwater diatomites of the early Miocene Foulden Maar Fossil-Lagerstätte in southern New Zealand.

Methods

We document 30 specimens of immature dipterans, mostly pupae, and compare their external morphology to extant aquatic Diptera. Based on the reconstructed paleoenvironment of Foulden Maar, we discuss taxonomic, ecological and taphonomic implications of this early Miocene fauna.

Results

Among Chironomidae, one pupal morphotype is attributed to Tanypodinae, one pupal morphotype and one larval morphotype are placed into Chironomus (Chironominae) and a further morphotype into Chironominae incertae sedis. Chaoboridae are represented by a pupal morphotype congeneric or very close to the extant Chaoborus, today globally distributed except for New Zealand. Additional immature specimens are likely larvae and puparia of brachyceran flies but cannot be identified to a narrower range. These finds document an aquatic dipteran fauna in New Zealand in the earliest Miocene and highlight Neogene extinction as a factor in shaping the extant Diptera fauna in New Zealand. Immature aquatic dipterans were a common and likely ecologically important component of the early Miocene Foulden Maar lake. Preservation of larvae and pupae may have been promoted by diatomaceous microbial mats and the light colour of the diatomite likely facilitated spotting of these minute fossils in the field.

Introduction

The isolation of New Zealand throughout geological time has resulted in an extremely unique and highly endemic flora and fauna of this southern land mass (Hennig, 1960; Brundin, 1966; Giribet & Boyer, 2010; Buckley, Krosch & Leschen, 2015). This uniqueness is particularly visible in the insect fauna, which contains many unusual radiations, relictual or depauperate lineages and unusual ecologies (Buckley, Krosch & Leschen, 2015; Macfarlane et al., 2010, and references therein). For instance, 91% of the ~3,200 species of true flies (Diptera) known from New Zealand are endemic, illustrating just how unique the history of this biota is Hennig (1960) and Macfarlane et al. (2010).

True flies play an important role for both ecosystems and humanity, due to their role as pollinators, decomposers of organic matter, parasitoids of agricultural pests and vectors of diseases (Marshall, 2012). One of the most intriguing phenomena in the New Zealand fauna is the absence of some highly mobile, widely distributed insect groups (Hennig, 1960). Such an absence is even more perplexing when these groups are present in Australia (Colless, 1986). One notable example is the group of phantom midges (Chaoboridae). Globally distributed, though species-poor, phantom midges are important ecosystem engineers and principal plankton predators in the lakes and ponds (Cook, 1956). Their absence from New Zealand freshwater ecosystems is apparently one of the causes for the unusual composition of the local plankton communities (Chapman & Green, 1987).

Another interesting case is the small, but highly endemic fauna of non-biting midges (Chironomidae) of New Zealand (Freeman, 1959; Hennig, 1960; Brundin, 1966; Macfarlane et al., 2010; Boothroyd & Forsyth, 2011). Non-biting midges are among the most widely distributed free-living insects in the world, with their representatives inhabiting depths of Baikal lake to 1,000 m, caves up to 980 m deep, mountain regions of the Himalayas over 5,600 m altitude as well as arctic wastes and continental Antarctica (Kohshima, 1984; Armitage, Cranston & Pinder, 1995; Ferrington, 2008; Andersen et al., 2016). Non-biting midges are important ecosystem engineers, influencing flux of organic matter and energy in both aquatic and terrestrial ecosystems, and they are important for carbon sequestering and water purification in freshwater and brackish ecosystems (Armitage, Cranston & Pinder, 1995; Gratton, Donaldson & Zanden, 2008; Baranov, Kvifte & Perkovsky, 2016; Herren et al., 2017). In New Zealand non-biting midges are represented by 149 species, 94% of them endemic (Macfarlane et al., 2010). This relatively high diversity and diverse ecological adaptations among Chironomidae relative to other dipteran groups suggest a long history of Chironomidae in New Zealand (Buckley, Krosch & Leschen, 2015). However, numerous ingroups of Chironomidae (“genera”—see comment on rankless taxonomy in the methods section) that are present in Australia are missing in New Zealand (Hennig, 1960; Ashe & O’Connor, 2009; Borkent, 2014). Despite numerous studies highlighting trans-Tasman biogeographic connections (Brundin, 1966; Cranston et al., 2010; Krosch et al., 2011, 2022), more paleontological data are needed to elucidate the biogeographic history of non-biting midges and, indeed, the rest of the New Zealand dipteran fauna.

Currently, the fossil record of New Zealand dipteran consists of a single larvae of Dilophus campbelli Harris, 1983 (Bibionidae) from the Eocene, as yet formally undescribed (but mentioned and to a degree figured) larvae, pupae and adults from two Miocene maar lakes (Kaulfuss et al., 2015, 2018a) and several formally undescribed (but mentioned and to a degree figured) representatives of Cecidomyiidae, Ceratopogonidae, Chironomidae and Mycetophilidae from late Oligocene–early Miocene ambers (Schmidt et al., 2018). Additionally, there are subfossil records such as larvae of Corynocera duffi Deevey, 1955 (Chironomidae) and associated non-brachyceran dipteran larva from Holocene swamp deposits (Deevey, 1955). With such a relatively sparse fossil record it is difficult to attain a good understanding of the biogeographic history of Diptera in New Zealand.

The Foulden Maar Fossil-Lagerstätte in southern New Zealand preserves a diverse flora and fauna that provides insights into the diversity and ecology of a rainforest/lake ecosystem in southern Zealandia in the early Miocene (23 mya) (Lee et al., 2016; Lee, Kaulfuss & Conran, 2022). Among the insect fauna from Foulden Maar are representatives of groups with aquatic or semi-aquatic life cycles, including dragonflies (Odonata, cf. Aeshnidae), caddisflies (Trichoptera), water scavenger beetles (Hydrophilidae), flies (Diptera) and alderflies (Sialidae), the latter now absent from New Zealand but present in Australia (Kaulfuss et al., 2015; Baranov et al., 2022a). Although the Foulden Maar fossil biota presents a rare opportunity for deciphering biogeographic connections of New Zealand’s flora and fauna (e.g., Lee, Kaulfuss & Conran, 2022), the aquatic insects of this Miocene lake ecosystem have been little studied.

Here, we describe new records of aquatic dipterans from the Foulden Maar fossil deposit and discuss these with regard to their paleoecology and to biogeographic patterns in the extant aquatic Diptera fauna of New Zealand. These records are allowing to further test biogeographical hypothesis regarding the extent of Oligocene submergence of New Zealand (Giribet & Boyer, 2010).

Geological setting

Foulden Maar is a partly eroded maar-diatreme volcano of the Dunedin Volcanic Group, an intracontinental volcanic field in the east Otago region, South Island of New Zealand, which was intermittently active between 25 and 9 mya (Scott et al., 2020). The Fossil Lagerstätte is located on private farmland east of the township Middlemarch. The principle fossiliferous lithology is a varved diatomite that accumulated in small maar lake over a period of at least 130,000 years. Sedimentological and geophysical investigations indicate that the maar lake occupied a semi-circular crater up to 2,400 m in diameter and 350 m deep, and was disconnected from streams and rivers by a rim of tephra deposited around the crater (Fox et al., 2015; Jones et al., 2017; Kaulfuss, 2017). Lateral continuous, undisturbed lamination of the diatomite in combination with the preservation of organic material and the absence of benthic organisms and bioturbation suggest meromictic conditions, with a mixed, well-oxygenated upper water body and an anoxic lower water column and lake floor unsuitable for aquatic life (Lindqvist & Lee, 2009). The slopes of the lake basin were generally very steep but swampy, shallow water edges during later stages of the lake’s existence are evidenced by pollen from bur reeds, bulrushes, flaxes, jointed rushes and sedges in the diatomite (Mildenhall et al., 2014a). Plant fossils in the diatomite include palynomorphs, leaves, flowers, fruits, seeds and bark from plants of a diverse, Lauraceae-dominated, warm-temperate to subtropical rainforest growing on fertile volcanic soils around the lake, and a pollen signal from regional forests dominated by Nothofagus (southern beech), Casuarina, podocarps and araucarians in the hinterland (Bannister, Conran & Lee, 2012; Mildenhall et al., 2014a; Lee et al., 2016; Lee, Kaulfuss & Conran, 2022). The fossil fauna recovered to date includes mygalomorph spiders (Selden & Kaulfuss, 2018), insects of the groups Odonata, Blattodea, Hemiptera, Megaloptera, Coleoptera, Hymenoptera, Trichoptera and Diptera (Kaulfuss et al., 2015; Baranov et al., 2022a), and larval to adult specimens of Galaxias effusus Lee, McDowall & Lindqvist, 2007 in the Southern Hemisphere family Galaxiidae (Teleostei) (Lee, McDowall & Lindqvist, 2007; Kaulfuss et al., 2020).

The age of the Foulden Maar biota is earliest Miocene (23 ma), Aquitanian, New Zealand local stages late Waitakian–early Otaian, pollen zones: latest uppermost Rhoipites waimumuensis Zone to lower early Proteacidites isopogiformis Zone (Mildenhall et al., 2014a).

Materials & methods

Studied material

The 30 immature individuals of flies described herein (Figs. 1–14) were collected in mining pit A (45.5269°S, 170.2191°E) at Foulden Maar (see Kaulfuss, 2017, Fig. 1C), which exposes a ca. 10 m thick succession of fossiliferous diatomite representing a depositional period of ca. 18,000 years. The site is registered as I43/f8503 in the New Zealand Fossil Record File (https://fred.org.nz/). Specimens were found randomly distributed throughout the stratigraphic section (no mass mortality layers were observed) and are always preserved on light-coloured diatomaceous sublaminae, which resulted from diatom blooms in warm seasons. No specimens were found in the dark, organic-rich sublaminae deposited in the cooler seasons.

Figure 1 Habitus of Tanypodinae pupae from Foulden Maar.

(A) Specimen OU46609 (191). (B) Specimen OU46626 (208). (C) Specimen OU46641 (223). (D) Specimen OU45541 (137). (E) Specimen OU44933 (28). (F) Specimen OU44930 (23).

Figure 2 Habitus of Tanypodinae pupa, specimen OU46626 (208).

(A) Habitus, unmarked. (B) Habitus, marked. Abbreviations: a1–a8, abdominal units 1 to 8; al, anal lobes; ct, cephalothoracic cuticle (cuticle covering head and thorax); fa, frontal apotome; th, thoracic horn (respiratory organ).

Figure 3 Details of Tanypodinae pupa, specimen OU46626 (208).

(A) Head and thorax. (B) Anal lobes. (C) Thoracic horn (respiratory organ), unmarked. (D) Thoracic horn, marked. Abbreviations: at, atrium; fc, felt chamber; lu, lumen of the horn; pp, plastron plate.

Figure 4 Habitus of Chironominae morphotype 1, cf. Chironomus, pupal exuviae.

(A) Specimen OU45543 (139). (B) Specimen OU47058 (252). (C) Specimen OU45553 (149). (D) Specimen OU46645 (227). (E) Specimen OU46627 (209). (F) Specimen OU46620 (202). (G) Specimen OU46654 (236). (H) Specimen OU47051 (245).

Figure 5 Habitus and details of Chironominae morphotype 1, cf. Chironomus, specimen OU47058 (252), pupal exuvium.

(A) Habitus, unmarked. (B) Habitus, marked. (C) Cephalothoracic cuticle. (D) Anal lobes. Abbreviations: a3–a6, abdominal units 3 to 6; ac, anal comb; al, anal lobes; th, thorax; wg, wings.

Figure 6 Habitus and details of Chironominae morphotype 1, cf. Chironomus, specimen OU45543 (139), pupal exuvium.

(A) Habitus. (B) Cephalothoracic cuticle, unmarked. (C) Cephalothoracic cuticle, marked. Abbreviations: an, antenna; fs, frontal setae.; ft, frontal tubercles.

Figure 7 Larval cuticle of Chironominae morphotype 2, cf. Chironomus, specimen OU47488 (268).

(A) Habitus, unmarked. (B) Habitus, marked. (C) Mouthparts, unmarked. (D) Mouthparts, marked. Abbreviations: ap, anterior parapods; ab, abdomen, hc, head capsule; md, mandible; mn, mentum; pm, premandibles; pp, posterior parapods; th, thorax.

Figure 8 Habitus of Chironomidae morphotype 1, cf. Chironominae, pupae.

(A and C) Specimen OU45549 (145). (B and D) Specimen OU46608 (190). (A) Habitus. (B) Habitus. (C) Thorax and head. (D) Thorax and head.

Figure 9 Habitus overview of Chaoboridae morhoptype 1, pupae.

(A) Specimen OU46653 (235). (B) Specimen OU46651 (233). (C) Specimen OU46642 (224).

Figure 10 Habitus of Chaoboridae morphotype 1, pupae.

(A) Specimen OU47487 (163). (B) Specimen OU46631 (213).

Figure 11 Thorax and head of Chaoboridae morphotype 1,specimen OU46642 (224).

(A) Thorax and head, unmarked. (B) Thorax and head, marked. Abbreviation: an, antenna; cl, clypeus; ey, eye; hl, hind leg; ht, hair tufts of abdomen; ml, mid leg; pd, pedicellus; wg, wing.

Figure 12 Thorax and head of Chaoboridae morphotype 1,specimen OU47487 (163).

(A) Thorax and head, unmarked. (B) Thorax and head, marked. Abbreviation: an, antenna; ey, eye; fl, front leg, mp, maxilar palpi; pd, pedicellus; sc, scutellum; wg, wing.

Figure 13 Habitus and details of Chaoboridae morphotype 1,specimen OU46653 (235).

(A) Habitus, unmarked. (B) Habitus, marked. (C) Hypopigium visible through the cuticle, unmarked. (D) Hypopigium visible through the cuticle, marked. Abbreviations: a3–a8, abdominal units 3 to 8; ae, aedeagus of the hypopigium;gc, gonocoxite;gs, gonostylus; hp, hypopigium; th, thorax; wg, wing.

Figure 14 Presumed brachyceran larvae and puparia from Foulden Maar.

(A) Specimen OU44996 (105). (B) Specimen OU44982 (91). (C) Specimen OU46644 (226). (D) Specimen OU46652 (234). (E) Specimen OU45559 (155). (F) Specimen OU44981 (90). (G) Specimen OU44944 (43). (H) Specimen OU46655 (237).

All individuals are pale to dark brown, strongly compressed (compacted) specimens, mostly preserved as part and counterpart, although often only the part exhibits useful morphological details, whereas the counterpart is a faint outline or impression only. One larva specimen, mounted on a glass slide (Fig. 7), was recovered from the carbonaceous residue of the diatomite. To this end, the silica component was dissolved in a plastic beaker with water and hydrofluoric acid, the residue washed and sieved, and the larva mounted in suspension onto a coverslip with a pipette. After removal of the water, the coverslip with the specimen was permanently mounted on a slide with an epoxide-based mounting medium (Petropoxy 154) (N Butterfield, 2022, personal communication). All specimens are stored in the Museum of the Geology Department, University of Otago (OU); identifiers provided below consist of an OU collection number followed by an original field number in brackets.

Imaging

Specimens were photographed with a Canon T3 camera attached to a Nikon SMZ1000 stereomicroscope. Wetting the specimens in ethanol improved the contrast between specimens and the diatomite matrix. The single slide-mounted specimen was imaged using a Keyence BZ-9000 fluorescence microscope with either 4×, 20×, 40× or 100× objectives. We have conducted observations using an emitted wavelength of 532 nm since it was the most compatible with the fluorescence capacities of the fossil specimen (Haug et al., 2011). Stacks of images were digitally computed to single in-focus images using CombineZP (GNU) or Photoshop CS5.1 software (Adobe Systems Inc., San Jose, CA, USA).

Morphological terminology and identification

In the course of our work we normally do not use Linnean ranks (‘rankless taxonomy’) (Baranov, Schädel & Haug, 2019; Baranov et al., 2022b; Haug et al., 2020). Ranks (or “categories” sensu Mayr, 1942, p. 102) are arbitrary constructs of the human-imposed structure that does not hold ‘comparative values’ (Mayr, 1942, p. 291, line 3). In our view such arbituary constructs do not contribute to facilitation of the understanding of phylogenetic relationships between the organisms, including both species and higher phylogenetic groups, instead we are using rankles hierarchy of the monophyletic groups (Haug et al., 2020).

The morphological terminology is based on Sæther (1980), Marshall et al. (2017) and Borkent & Sinclair (2017), and specifically follows Borkent (2012) for culicomorphan pupae anatomy. In this article we describe morphotypes, i.e., distinct morphological groups of organisms. Members of each morphotype are here assumed to represent the same species, although this is often impossible to ascertain for the fossils. Most of the fossils dealt with herein are pupal exuviae (integument left after the eclosion of the adult fly) but for convenience we treat all pupae fossils and their integuments as “pupae”.

Specimens were identified using the keys provided by Wiederholm (1986), Langton (1991), Sæther (1970, 1997), Roback (1971), Forsyth (1971), Cook (1956), Colless (1983) and Winterbourn, Gregson & Dolphin (1989).

Results

Systematic paleontology

DIPTERA Linnaeus, 1758

CHIRONOMIDAE Newman, 1838

TANYPODINAE Thienemann & Zavřel, 1916

Material: specimens OU46609 (191), OU46626 (208), OU45541 (137), OU44933 (28), OU44930 (23), OU46641 (223) (Figs. 1–3).

Pupa. Habitus. Medium-sized. All specimens preserved in dorso-ventral aspect, thus precluding detailed observation of head and legs. Body length 4.5–5.4 mm (n = 2); abdomen length 2.9–3.7 mm (n = 2); length of thorax 1.6–1.7 mm (n = 2), body differentiated into presumably 20 segments, ocular segment plus 19 post-ocular segments (Figs. 2A, 2B); anterior part of body composed of head and thorax, visible as single semi-circular structure; thorax with wings and ambulatory appendages (legs) (Figs. 1A–1F, 2A, 2B); ocular segment and post-ocular segments 1–5 (presumably) forming a distinct capsule (head capsule); mouthparts located ventrally and thus not available for observation (Figs. 1A–1F). Pupal cuticle of the head with prominent frontal apotome (frontal protrusion of the head), apparently bearing a pair of strong frontal setae (only right one is visible), sitting on the short tubercle (Fig. 3A).

Thoracic segments forming a single semi-globose structure closely enveloping the head of the pupa. Prothorax bears thoracic horns (modified first spiracle) (Figs. 2A, 2B, 3A, 3C, 3D), these mostly cylindrical, 4.7 times as long as wide, with a total length of 400 µm (n = 1, only one horn was preserved in a good condition) (Figs. 3C, 3D). Widest point of thoracic horn at base of plastron plate (surface for retention of the air film, providing a gas exchange interface), overall shape of the horn tapers slightly towards the base (Figs. 3C, 3D). Plastron and atrium connection area light colored. Plastron plate ovoid, 90 µm long and almost 90 µm wide. (Figs. 3C, 3D). Thoracic horn with apical papilla (Figs. 3A, 3C). Thorax is very wrinkly, bearing no distinct sclerotized protrusions (i.e., thoracic combs, sensu Saether, 1980). Wings and their cuticular sheath barely visible, hidden under the body, only points of the wing attachment to the mesothorax discernible (Figs. 2A, 2B).

Abdomen (posterior trunk). Abdominal cuticle extremely wrinkly. First unit of abdomen with very strongly pigmented scar (Figs. 1A–1F, 2A, 2B). Setae of abdominal units not preserved, but sturdy tecae of some of them can be seen on some abdominal tergites. Abdominal tergites 3 and 4 bearing tecae of five dorsal setae each, abdominal tergite 5 with at least one pair of tecae of dorsal setae (Fig. 2A). Traces of lateral setae on abdominal units 2–6 not discernible. Abdominal unit 8 with traces of bases of some lateral setae. Trunk end (abdominal unit 9 plus remnants of abdominal unit 10) bears anal lobes (paddles) (Fig. 2A), these semi-circular, broadly rounded, 1.6 times longer than wide. Anal lobes bearing a fringe of spine-like, short setae, present in the distal part of the lobes only (Figs. 2A, 2B, 3D).

Taxonomic attribution. These specimens are representatives of Chironomidae based on the following combination of pupal characters: thoracic horn with strong plastron plate; strongly sclerotized arches from the anterior parts of the abdominal tergites; terminus of trunk without articulated terminal paddles (Figs. 2A, 3D). Within Chironomidae, this morphotype falls within the Tanypodinae based on the following combination of characters: thoracic horn well developed, with prominent plastron; anal lobes rounded, with fringe of the setae along the outer edge of the anal lobes, anal lobes completely confluent and with overlapping inner edges (Fig. 3A) (Fittkau & Murray, 1986). It is difficult to place this morphotype within Tanypodinae, due to many characters being un-observable. However, the structure of the thoracic horns, in particular light colored connection between plastron and atrium and presence of apical papilla and overall shape (broadly rounded) of the anal lobes are pointing to the Procladiini or Anatopyiini, with Procladius being a particularly close match, due to spine like setae present only on the distal part of the anal lobes (Fittkau & Murray, 1986).

Remarks. The are no extant representatives of Tanypodinae with morphologically similar pupae in New Zealand, but Forsyth (1971) reported a number of species of “Anatopynia”, of which some share a certain degree of similarity with this fossil morphotype. In particular, “Anatopynia” anatarctica (Hudson, 1892) shares with the new fossil morphotype the broad, rounded shape of the anal lobes and the general structure of the thoracic horn (Forsyth, 1971: Fig. 2). However, “A.” antarctica has prominent distal protruding points on the anal lobes, which are absent from the new morphotype from Foulden Maar. Additionally, the inner edges of the anal lobes are overlapping in the new morphotype (Fig. 3A), which is characteristic for representatives of Coelotanypus, but not for any of the representatives of “Anatopynia” recorded from New Zealand (Forsyth, 1971; Fittkau & Murray, 1986).

CHIRONOMINAE Macquart, 1838

CHIRONOMINI Macquart, 1838

cf. Chironomus Meigen, 1803

morphotype 1 (pupae)

Material: OU47058 (252), OU46654 (236), OU46645 (227), OU46627 (209), OU46625 (207), OU46620 (202), OU45553 (149), OU45543 (139), OU47051 (245) (Figs. 4–6).

Pupa. Habitus. Medium-sized, coma-shaped (in lateral aspect). Most specimens preserved in dorso-ventral aspect, thus precluding detailed observation of head and legs. Body length 4.7–6.2 mm (mean = 5.6 mm, sd = 430 µm) (n = 8); abdomen length 3.6–4.5 mm (mean = 4.2 mm, sd = 355 µm) (n = 8); length of thorax 1.3–2.3 mm (mean = 1.6 mm, sd = 330 µm) (n = 8). Body differentiated into presumably 20 segments, ocular segment plus 19 post-ocular segments (Figs. 4A–4H, 5A–5H, 6A–6E); anterior part of body composed of head and thorax, visible as single semi-circular structure; thorax bears wings and ambulatory appendages (legs) (Figs. 6A, 6B); ocular segment and post-ocular segments 1–5 (presumably) forming a distinct capsule (head capsule); mouthparts located ventrally and thus not discernible (Figs. 5A–5H, 6A, 6B). Pupal cuticle of head with prominent frontal apotome (frontal protrusion of the head). Frontal apotome bears a pair of strongly curved, conical cephalic tubercles with a pair of strong frontal setae (Figs. 6A–6C).

Thorax with three pairs of ambulatory appendages (fore-, mid- and hindlegs) on the pro-, meso-, and metathorax, respectively. Thoracic segments forming a single semi-globose structure closely enveloping the head of the pupa. Anterolateral and anteromedian tubercles absent. Forelegs folded around dorsal side of wing (Figs. 6A, 6B). No traces of the thoracic horns were found, but specimen OU45543 (139) shows the presence of the tracheal scar (place where trachea is passing through the thorax cuticle into the thoracic horns). Mesothorax with a pair of wings and a pair of ambulatory appendages (midlegs). Midlegs situated medially to forelegs, looping around wing, distal part of the loop lying on the abdomen, beyond the distal end of the wing. (Figs. 6A, 6B). Metathorax with a pair of ambulatory appendages (hindlegs); halteres not discernible. Hindlegs almost entirely hidden behind wings (Figs. 6A, 6B).

Abdomen (posterior trunk). Made up of nine visible abdominal units. Tergal armament most complete and best preserved in specimens OU46654 (236), OU47051 (245) and OU47058 (252) (Figs. 5B, 5G, 5H, 6A, 6B). Abdominal tergites unmodified, without plates or spine mounds. Abdominal tergite 1 bare. Abdominal tergite 2 with fine shagreen pattern and continuous row of hooks at posterior edge (Figs. 6A, 6B). Abdominal tergites 3–5 with uniform shagreen and strong oval patch of longer, dark spines located on the median line of tergite, touching the posterior edge of the segment. Tergite 2 with hook row narrowly interrupted or continuous, but definitely without a wide gap. Tergites 2–4 without anterior band of shagreen. Tergite 6 apparently mostly bare, without visible shagreen, bearing similar medio-posterior patch of strong, dark spines (Figs. 6A, 6B). Tergites 7 and 8 mostly bare, without visible shagreen (Figs. 6A, 6B). Abdominal unit 8 bearing two strong anal combs postero-lateraly (Figs. 6D, 6E), these made up of four strong spines, the outermost being the longest and the others getting shorter towards the innermost spine. Anal lobes semi-circular, with strong fringe of at least 50 setae (difficult to count) on each lobe (Fig. 6D), without any processes or modified setae. Genital sacs of the male without spines. No other setae are preserved on the abdomen of this morphotype. In specimen OU46645 (227), a part of a male hypopigium is visible through the cuticle of the genital sack. Strong, blunt anal point visible, alongside a long, curving gonostyle, joined to a gonocoxite (Fig. 5D).

Taxonomic attribution. Specimens of this morphotype are representatives of Chironomidae based on the following combination of pupal characters: strongly sclerotized arches from the anterior parts of the abdominal tergites; terminus of trunk without articulated terminal paddles (Figs. 5A–5H, 6A, 6D). Within Chironomidae, this morphotype can be interpreted as an ingroup of Chironominae because of the diagnostic combination of characters: abdominal tergite 8 with a strong anal comb, anal lobes with a well-developed fringe of setae, gonostylus and gonocoxite conjoined rigidly, no articulation visible (Fig. 5D) (Wiederholm, 1989). This morphotype falls within Chironomini based on the following combination of characters: tergites 2–6 without paired patches of shagreen, bearing rectangular fields of shagreen instead, wing sheath without “nose” protrusion on the distal end (Wiederholm, 1989). Within Chironomini, this morphotype appears to be a representative of Chironomus based on the following combination of characters: cephalic tubercles present but without apical field of spines, frontal warts absent, frontal setae present, anterolateral and anteromedian tubercles absent; tergites of the abdomen bearing large but unmodified fields of shagreen, tergites 2–4 without anterior band of shagreen; hook row of tergite 2 without a wide gap, abdominal tergites without anteromedian toothed lobe, abdominal tergite 6 without posteromedian mound of spines, anal lobes without brushes of dark setae and without long spines; anal lobe bearing uniform fringe of setae, forked process, genital sacs of males without spines, anal comb in form of elongated spine with three points (Wiederholm, 1989, Forsyth & Martin, 2014).

CHIRONOMINI Macquart, 1838

cf. Chironomus Meigen, 1803

morphotype 2 (larva)

Material: OU47488 (268) (Fig. 7).

Larva. Habitus. A single specimen mounted on a glass slide (Figs. 7A–7D). Larva with well-preserved body and head capsule cuticle, with many microscopic details such as structure of submentum or mandible apparent. Head capsule well developed, with complete sclerotization and overall non-retractable. Ocular segment and post-ocular segments 1–5 (presumably) forming a distinct capsule (head capsule). Head capsule without conspicuous labral fans and well-developed epipharyngeal complex (of which only premandibles are visible). Head capsule bears pair of mandibles. Mandibles with pronounced apical tooth and three internal teeth (Fig. 7C, 7D). Premandibles apparently with more than three apical teeth. Mentum (part of labium) well pronounced, with seven pairs of lateral teeth, with first part of lateral teeth about four times shorter than second pair, with wide gap between them.

Head and thorax not conjoined together; no suction discs at the abdomen; abdominal cuticle well preserved, but without visible setae; respiratory system lacking developed trachea or external spiracles (apneustic type); thorax segments well distinguishable; prothorax and abdominal unit 9 with paired parapods; abdominal units 1 and 2 without parapods; group of strong, downward pointing preanal setae absent on the trunk end.

Taxonomic attribution. This morphotype falls within Chironomidae based on the combination of the following characters: larva with well-developed, complete and non-retractable head capsule; head capsule without conspicuous labral fans and well-developed epipharyngeal complex; head and thorax not conjoined together; no suction discs at the abdomen; respiratory system lacking developed trachea or external spiracles (apneustic type); thorax segments well distinguishable; prothorax and abdominal unit 9 with paired parapods; abdominal units 1 and 2 without parapods; group of strong, downward pointing preanal setae absent on the trunk end (Ekrem et al., 2018). Within Chironomidae, this larva falls within Chironominae based on the following combination of characters: antenna not retractable, non-annulate, labrum without row of the overlapping lamellae, premandible present, submentum with 15 teeth in three distinct groups, symmetrically distributed on submentum (Wiederholm, 1983; Cranston, 2019).

The preservation of the head capsule is not conducive for further identification of the specimen but the general habitus is highly reminiscent of that of Chironomus Meigen, currently represented in New Zealand by at least six species (Boothroyd & Forsyth, 2011). We propose affinity of this morphotype to Chironomus based on the following combination of the characters: mandible with three inner teeth, premandible with seemingly more than three teeth, central tooth of the mentum trifid (Wiederholm, 1983; Cranston, 2019). It is important to note that with the characters available it is impossible to positively differentiate this morphotype from Goeldichironomus Fittkau, 1965.

cf. CHIRONOMINAE Macquart, 1838

Chironomidae morphotype 1

Material: OU45549 (145), OU46608 (190) (Fig. 8).

Pupa. Both specimens are too poorly preserved for a detailed description (Figs. 8A–8D).

Taxonomic attribution. These two specimens of pupae belong to a separate morphotype, which is difficult to place due to the missing characters of the distal end of the abdomen, yet their overall habitus is highly reminiscent of that of Chironomidae. Defining feature of this morphotype is a strong, protruding spine on the posterior edge of tergite 2 (Figs. 8A–8D). Since value of leg ratio (LR = length of tarsomere one divided by the length of tibia) is >1, it is probable that this morphotype belongs to Chironomidae (Wiederholm, 1989).

CHAOBORIDAE Edwards, 1912

Chaoboridae morphotype 1

Material: Specimens OU47487 (163), OU46642 (224), OU46631 (213), OU46651 (233), OU46653 (235) (Figs. 9–13).

Pupa. Habitus. Medium-sized, coma-shaped (in lateral aspect). Body length 5.2–6.0 mm (n = 3, mean = 5.5 mm, sd = 490 µm); abdomen length 3.8–4.3 mm (n = 3, mean = 4.0 mm, sd = 320 µm); length of thorax 1.3–18.8 mm (n = 4, mean = 1.5 mm, sd = 220 µm). Body differentiated into presumably 20 segments, ocular segment plus 19 post-ocular segments (Figs. 13A, 13B); anterior part of body composed of head and thorax, visible as a single globose structure; thorax bears wings and ambulatory appendages (legs) (Figs. 11A, 11B, 12A, 12B, 13A, 13B); mouthparts located ventrally and short, ending before attachment of first ambulatory appendages (forelegs) (Figs. 11A, 11B, 12A, 12B, 13A, 13B). Ocular segment recognizable by its appendage derivative, clypeo-labral complex, and a pair of large compound eyes. Labrum and clypeus present, but their shape obscured by deformation and preservation in lateral aspect (Figs. 11A, 11B, 12A, 12B). Antennae curved around the head, ending beneath the head at about mid-length to 0.8 of the length of the wings. Antennae attached to massive, rounded pedicellus (antennal element 2) (Figs. 11A, 11B, 12A, 12B). Maxilla recognizable by maxillary palpus, palpi poorly preserved. Post-ocular segment 5 recognizable by its appendages, forming the labium [conjoined left and right maxillae]. Labium mostly obscured in all specimens, with no details visible (Figs. 11A, 11B, 12A, 12B). Thorax bears three pairs of ambulatory appendages (fore-, mid- and hindlegs) on the pro-, meso-, and metathorax, respectively. Thoracic segments forming a single semi-globose structure, closely enveloping the head of the pupa. Ambulatory appendages of the thorax folded around and under the wings (Figs. 11A, 11B, 12A, 12B). Prothorax bears thoracic horns (modified first spiracle). Thoracic horns (respiratory organs) absent in all specimens. Prothorax bears first thoracic appendages (forelegs). Forelegs running posteriorly, upwards anteriorly to upper edge of eye and then downward to the apical edge of wing (Figs. 11A, 11B, 12A, 12B, 13A, 13B). Mesothorax bears a pair of wings and a pair of ambulatory appendages (midlegs). Midlegs situated medially to foreleg, looping around the wing, distal part of the loop lying on the abdomen, beyond the distal end of the wing. Distal parts of midlegs loop again under the wing (Figs. 11A, 11B, 12A, 12B, 13A, 13B). Hindlegs almost entirely hidden behind coxae of the fore- and midlegs and wings (Figs. 11A, 11B, 12A, 12B, 13A, 13B).

Abdomen (posterior trunk). Abdominal units 1–8 with setae on the pharate adult tergites visible through the pupal cuticle. Setae radiating from the median line of the abdomen diagonally, so as to form pointed bundleds of setae at the posterio-lateral part of tergites 1–8 (Figs. 11A, 11B, 13A, 13B). Trunk end (abdominal unit 9 plus remnants of abdominal unit 10) bears hypopigium (male genitalia) (only visible on specimen OU46653 (235)). Hypopigium consists of paired gonocoxites (II) of abdominal unit 9 and paired gonostyli articulated at the distal end of gonocoxites (Figs. 13C, 13D). Gonocoxites ca. 250 µm long, gonostyli ca. 430 µm. Gonostyli straight, ending with short, rounded apical setae (megaseta) (Figs. 13C, 13D). Gonocoxites densly covered with strong setae. No traces of anal lobes present on the trunk end (Figs. 13C, 13D).

Taxonomic attribution. We interpret this new morphotype as an ingroup of Chaoboridae based on the specific combination of the following characters: Pupa: mouthparts short, not reaching beyond coxae of anterior legs; bundles of diagonally oriented setae on tergites of the abdominal units (the latter character is a autapomorphy of Chaoboridae) (Figs. 11A, 11B, 13A, 13B) (Borkent & Grimaldi, 2004; Borkent, 2012). Some characters of the adult male were available for examination through the cuticle of the pharate adult (inside the pupa) in specimen OU46653 (235) (Figs. 13C, 13D). However, the poor preservation does not allow for a closer taxonomic attribution as characters of the adult legs and wings are not discernable in this specimen. All the available specimens of Chaoboridae from Foulden Maar are missing anal lobes and thoracic horns. These characters are crucial for diagnosis and taxonomic attribution of representatives of Chaoboridae (Cook, 1956; Sæther, 1970). Therefore, we cannot attribute this new morphotype to any ingroup of Chaoboridae, although the general shape of the pupa and shape of the hypopigium are very similar to pupae of Chaoborus Lichtenstein (Cook, 1956; Sæther, 1970, 1997; Borkent, 2012). We thus suggest that the Chaoboridae morphotype from Foulden Maar is either a representative of Chaoborus or closely related to it. To further validate this assumption, we will require additional material with preserved anal lobes and thoracic horns. The loss of anal paddles in pupae from the finely laminated Foulden Maar diatomite is likely due to the fragility of the “paddle” attachments (anal lobe). A similar preservation is present at the Eocene Kishenehn Formation, USA, and the Miocene McGrath Flats Formation, Australia, where otherwise exquisitely preserved Chaoboridae pupae are frequently missing their anal “paddles” (anal lobes) (Baranov et al., 2022b; McCurry et al., 2022).

BRACHYCERA (?)

Material: OU44944 (43), OU44981 (90), OU44982 (91), OU44996 (105), OU45559 (155), OU46644 (226), OU46652 (234), OU46655 (237) (Fig. 14).

The examined material contains eight specimens, seemingly immatures of a holometabolan; all share similar dark-brown or reddish-brown colours (Figs. 14A–14H). All specimens have at least 11 body units and some have their supposed tergites split in half, giving an impression of the post-eclosion pupal exuvia (Figs. 14C, 14D, 14H). Tergites are split in the middle. Some specimens have strong setae on the edges of the tergites (Figs. 14B, 14D, 14F). Specimens OU44944 (43) and OU44996 (105) are more of a spindle-shape and have of an overall appearance of dipteran larvae, rather than pupal exuvia or puparia (Figs. 14A, 14G). Unfortunately, there are no diagnostic characters allowing for a closer taxonomic placement of these specimens. Based on the habitus similarity, we hypothesize that these specimens represent larvae and puparia (larval last stage exuvia, covering a pupa of brachyceran flies) of brachyceran flies, likely representatives of Cyclorrhapha (i.e., see Ferrar, 1987, vol 2 Fig. 6.155 (p. 568)).

Discussion

Paleontological significance and paleoecology

Fossil deposits in southern New Zealand document the presence of various lentic habitats and associated freshwater faunas during the early and middle Miocene (e.g., Douglas, 1986; Pole, Douglas & Mason, 2003; Lee et al., 2016; Kaulfuss et al., 2018a, 2020). Previously reported fossils of the groups Odonata, Megaloptera, Coleoptera, Diptera and Trichoptera provide a glimpse of the aquatic insect fauna in these paleo-habitats (Kaulfuss et al., 2015, 2018a, 2018b; Schmidt et al., 2018; Baranov et al., 2022a), but the diversity, biogeography and ecological role of individual groups remain to be explored in detail. Our study gives insights into the aquatic dipteran fauna in a small, isolated maar lake in southern New Zealand in the earliest Miocene (23 mya), shortly after maximum marine inundation of most land area at ~25 mya. The larvae and pupae described here from the Foulden Maar diatomite indicate the presence of a dipteran fauna consistent in its taxonomic richness with the diversity of merolimnic flies recorded from other Paleogene and Neogene lacustrine deposits: Kishenehn Formation, Eocene, USA, eight morphotypes (Baranov et al., 2022b), McGrath Flats, Miocene, Australia, five morphotypes (McCurry et al., 2022), Messel, Eocene, Germany, eight morphotypes (Paleobiology Database, 2023), Randecker Maar, Miocene, Germany, one morphotype (Paleobiology Database, 2023).

The non-biting midges (Chironomidae) from Foulden Maar include Tanypodinae, one pupal and one larval morphotype in Chironomini (Chironominae), both resembling Chironomus, and a further chironomid morphotype of uncertain identity, possibly in Chironominae. The only non-biting midge fossils previously reported from New Zealand are four adult specimens of Bryophaenocladius Thienemann (Orthocladiinae) from late Oligocene amber (Schmidt et al., 2018). Phantom midges (Chaoboridae) from Foulden Maar are represented by a pupal morphotype congeneric or closely related to the widespread and speciose extant group Chaoborus. These pupae are the first fossil record of phantom midges from New Zealand. Intriguingly, phantom midges are absent in the extant fauna of New Zealand (see below). Adult life stages of non-biting and phantom midges have not yet been found in the Foulden Maar diatomite. The presence of immature aquatic brachycerans in the Foulden Maar paleo-lake is documented by eight larvae and puparia of uncertain systematic position. Rare isolated wings of adult brachyceran flies (either representatives of Muscidae or Acalyptrata) have previously been reported from the fossil site (Kaulfuss et al., 2015), but affiliation of these with any of the immature aquatic specimens cannot be established due to incomplete preservation.

The comparatively small sample of identifiable insects (n = 253) from Foulden Maar includes a relatively high proportion of immature aquatic dipterans (16%), suggesting that these life stages were a common component in this limnic paleo-ecosystem. Of these, non-biting pupae are most common (63% of immature dipterans), followed by the brachyceran-type (20%) and by phantom midges (17%). Together with other Crustacean forms, they likely provided a food source for fish, in particular for Galaxias effusus, which is commonly found as larvae, juvenile and adult specimens in the diatomite (Lee, McDowall & Lindqvist, 2007; Kaulfuss et al., 2020). The most common type of coprolite at Foulden Maar is most likely derived from Galaxias effusus and consists of mineral grains, plant material and common euarthropodan fragments (Kaulfuss, 2013). The latter are yet to be studied in detail and it is currently not known if non-biting or phantom midge remains are present in these coprolites. In any case, the occurrence of euarthropodan remains in these coprolites conforms to a diet of aquatic and terrestrial insects observed in most extant Galaxias species in New Zealand (McDowall, 2010). Immature dipterans are also commonly preyed on by various groups of aquatic eurthropodan animals such as water mites, dragonflies and damselflies larvae, aquatic bugs and beetles (e.g., Armitage, Cranston & Pinder, 1995; Martin & Gerecke, 2009; Ferrington, 2008; Klecka & Boukal, 2012). Although this probably was the case in the Foulden Maar lake, the available fossil data are insufficient for establishing specific predator-prey relations.

Assuming the ecology of the Chaoboridae midges from Foulden Maar concurs with that of extant relatives (e.g., Macdonald, 1956; Rudstam, 2009; Hare & Carter, 1986), its larvae may have been abundant in the pelagic and littoral zones of the maar lake, feeding on small eucrustaceans (e.g., of the groups Copepoda and Cladocera), benthic organisms and dipteran and other insect larvae, and possibly also ingesting readily available phytoplankton. As in extant species of Chaoborus, late stage larvae (3rd and 4th instar), probably exhibited diurnal vertical migration, preying in the epilimnion at night and migrating into deeper, oxygen-depleted zones of the monimolimnion or sediment to avoid fish predators during day (Macdonald, 1956; Hare & Carter, 1986).

Chironominae morphotype 2 is closely resembling larvae of the extant species of Chironomus and probably had a similar ecology, inhabiting soft sediments and relying on acquisition of food by bioirrigation, pumping water containing organic particles through their burrows in the sediment (Hamburger, Dall & Lindegaard, 1994). Tanypodinae is represented at Foulden Maar by specimens of probable Procladiini. Extant representatives inhabit fine-grained sediments, where they rely heavily on bioirrigation for food acquisition (Boesel, 1974; Matisoff & Wang, 1998). The larvae also prey on other bioirrigating animals, such as worms of the group Tubificidae (Soster & McCall, 1989) and are capable of utilising other sources such as suspended organic particles and detritus (Boesel, 1974; Matisoff & Wang, 1998).

The immature dipterans at Foulden Maar are primarily exuviae of pupae, left after eclosion of the adult, terrestrial stage. Pupae of both non-biting and phantom midges are short-lived, lasting only for several hours to several days, but are ecologically important in providing large amounts of organic matter in pulses to the higher order consumers in lakes and rivers (Lehman et al., 1998; Wagner, Volkmann & Dettinger-Klemm, 2012).

Biogeographical implications

Chironomidae

The major ingroups (“subfamilies”) of Chironomidae originated between the mid-Triassic and the early Cretaceous (Cranston, Hardy & Morse, 2012) when Zealandia was connected to the great Southern landmass of Gondwana. Morphology-based studies have recognised trans-Antartic relationships among Southern non-biting midges and argued for vicariance origin via break-up of Gondwana. For instance, Brundin (1965) detected an “old Antarctic element of Southern lands” for some groups of Southern temperate non-biting midges and suggested a Mesozoic orogenic belt corresponding to present New Zealand, Western Antarctica and Western Patagonia as centre of evolution. Divergence dates from phylogenetic studies correlate with a vicariance origin for some South American and Australian groups but indicate a more complex history for non-biting midges of New Zealand (Cranston et al., 2010; Krosch & Cranston, 2013; Krosch et al., 2017). Some nodes separating Australian and New Zealand Tanypodinae and other ingroups of Chironomidae have been dated at ca. 50 mya and are indicative of Eocene dispersal, possibly via an archipelago connection provided by the Lord Howe Rise and/or the Norfolk Ridge northwest of New Zealand (Krosch et al., 2017; Krosch & Cranston, 2013). These reconstructed ancestral nodes post-date the separation of New Zealand from Gondwana at ~80 mya but they pre-date the Oligocene “drowning”, a period of near-complete (or complete, according to some authors) submergence of the New Zealand landmass 25–23 mya (Cooper & Cooper, 1995; Mildenhall et al., 2014b; Kamp, Vincent & Tayler, 2015; Wallis & Jorge, 2018). The non-biting midge fauna described here from a 23 million-year-old freshwater lake is contemporaneous or slightly younger than the late Oligocene maximum marine transgression. Together with specimens of Bryophaenocladius reported from late Oligocene (Duntroonian, 27.3–25.2 my) New Zealand amber (Schmidt et al., 2018), they argue for the presence of freshwater habitats and associated non-biting faunas during near-complete Oligocene inundation. For the other non-biting midges, which could not be identified to a narrower range, from Foulden Maar it impossible to establish closer biogeographic relationships.

Chaoboridae

Previously, Corethrella novaezealandiae Tonnoir, 1927 had been reported as the sole phantom midge from New Zealand, but this species is now generally accepted as a representative of Corethrellidae (frog biting midges; = sister group to Chaoboridae + Culicidae) (Wood & Borkent, 1989). Despite their (otherwise) cosmopolitan distribution no phantom midges are present in the New Zealand fauna (Chapman & Green, 1987; Borkent, 2014). The neighbouring landmass of Australia has a small, but distinct fauna of Chaoboridae with seven extant species (Colless, 1986; Borkent, 2014) and fossil records from the Early Cretaceous and the Miocene (Jell & Duncan, 1986; McCurry et al., 2022).

Chaoboridae is an ancient group of Diptera with the oldest record dating back to the Triassic (Ladinian–Carnian, Madygen Formation in Kyrgyzstan; Lukashevich, 2022). Chaoboridae has a rich fossil record, including some of 41 species, which is almost exclusively from the Holarctic realm and suggests an East Asian origin of the group (Kalugina & Kovalev, 1985; Ogawa, 2007; Borkent, 2014). More complete fossil material of the Chaoborus-like fossils from Foulden Maar is needed for detecting possible relationships to other, extant or extinct species. For now, these fossils demonstrate that phantom midges were present in New Zealand freshwater habitats by the earliest Miocene. Yet, it is unclear whether this reflects an ancient vicariance origin or pre-Miocene arrival from Australia or elsewhere. Likewise, there is no clear indication for a possible cause of the post-early Miocene extinction of phantom midges in New Zealand. Phantom midges are generally very adaptable animals with larvae successfully developing in a broad range of stagnant and slowly flowing water bodies, of which there is currently no lack in New Zealand (Colless, 1983, 1986; Chapman & Green, 1987) and very likely has not been since at least the Eocene and probably since its separation from Gondwana in the Cretaceous (Buckley, Krosch & Leschen, 2015, and references therein on pp. 7–8). It stands to reason that there is a great potential for phantom midges to thrive in aquatic habitats of New Zealand, and their absence from the region cannot be explained by a lack of suitable habitats. The find of Chaoboridae at the early Miocene Foulden Maar might prove important for the critical re-evaluation of the impact of Oligocene submergence on the New Zealand biota.

Taphonomy

Immatures of the groups Chironomidae and Chaoboridae may be common in Mesozoic and Cenozoic lacustrine settings, concurring with their aquatic lifestyle (e.g., Sinichenkova & Zherikhin, 1996; Johnston & Borkent, 1998; Baranov et al., 2022b; McCurry et al., 2022). However, in Cenozoic maar lakes with well-documented insect faunas (>4,000 studied specimens) immature aquatic midges (and other groups of aquatic insects) are typically absent or only present in small numbers. This has been attributed to supposed unfavourable limnic conditions in these small but deep lakes or to taphonomic biases, which favour the preservation of larger and more compact adult insects in the profundal sediments that are usually excavated for fossils (Lutz, 1991, 1997; Wedmann, 2000; Wedmann, Poschmann & Hörnschemeyer, 2010; Wedmann et al., 2018; Wappler, 2003). At the Eocene Eckfeld Maar, for instance, the study of ~4,600 insects yielded only two pupae and several larval cases of Chironomidae, and no chaoborid midges were found (Wappler, 2003). No aquatic stages of non-biting or phantom midges were reported from the rich insect faunas of the Oligocene Enspel Maar (Wedmann, Poschmann & Hörnschemeyer, 2010) and the Paleocene maar of Menat (Wedmann et al., 2018). Immature non-biting and phantom midge body fossils are also absent in the ‘oilshales’ of the Messel Maar, although their remains are frequently encountered in fish coprolites, indicating ecologically important populations in this Eocene maar lake (Richter & Baszio, 2001; Richter & Wedmann, 2005; Wedmann & Richter, 2007). Exceptions appear to be the Miocene maar lakes at Öhningen and Randeck, where aquatic insects including immature Diptera have been reported as being relatively common (Heer, 1865; Joachim, 2010).

At Foulden Maar, the proportion of immature aquatic dipterans at is comparatively high, which may reflect preferential conditions for the preservation of small aquatic insects in this Miocene lake. Several taphonomic studies have highlighted the role of diatom mats in the exceptional preservation of fossil biotas (Harding & Chant, 2000; O’Brien, Meyer & Harding, 2008; Iniesto et al., 2016; Olcott et al., 2022). Diatomaceous microbial mats may facilitate fossilisation by entrapping and transporting macrobiota through the water column quickly and by stabilising the sediment surface on the sea/lake floor (Harding & Chant, 2000; Olcott et al., 2022). Additionally, extracellular polymeric substances secreted by diatomaceous mats may form a chemical microenvironment (microbial sarcophagus) that enables fossilisation by delaying decay and inducing biomineralization of euarthropods, vertebrates and plants (Iniesto et al., 2016; O’Brien, Meyer & Harding, 2008; Olcott et al., 2022). At Foulden Maar, all euarthropodan fossils are preserved in laterally continuous, light-coloured laminae essentially composed of siliceous diatom frustules. The dominating species is Encyonema jordaniforme Krammer, 1997, a pennate and likely mucilaginous diatom that flourished in the upper water column of lake Foulden and formed annual diatom blooms over the lake’s estimated life span of 130,000 years (Harper et al., 2019). Centric diatoms are present as minor constituents in the lake sediment (Kaulfuss, 2017). Although taphonomic processes for the Foulden Maar biota are yet to be studied in detail, it is likely that diatomaceous mats essentially composed of E. jordaniforme might have provided a taphonomic pathway for the preservation of small aquatic larvae/pupae and other macrobiota in the diatomite.

Sediment colour might also have an impact on the apparent proportion of aquatic dipterans and other small euarthropods in fossil deposits. The Cenozoic maars at Eckfeld, Messel, Enspel and Menat, where immature non-biting or phantom midges are absent or rare, are primarily made of dark, organic-rich clay and mudstones, which makes spotting small fossils of similarly dark colour difficult. For Messel Maar, Wedmann & Richter (2007) argued that phantom midge larvae are likely present in the sediment, but their weakly sclerotised, translucent body cannot be seen in the organic shales. At Foulden Maar, the light-coloured (white to beige) diatomaceous laminae exhibit a pronounced colour contrast to embedded, typically brown or black fossil organisms. This contrast likely facilitates spotting of small euarthropods such as dipteran pupae in the field, and it might be one factor for the relatively high abundance of aquatic dipterans relative to most other maar-type Lagerstätten. The other two other Cenozoic maar lakes with a relatively high abundance of immature aquatic Diptera also consist of, or at least include light-coloured lithologies. At the mid-Miocene Öhningen Maar, many insects were recovered from white limestones (“Weißer Schieferstein”) and light-grey marlstones (“Kesselstein”) (Rasser et al., 2023). Similarly, the main insect-bearing lithologies with immature aquatic dipterans at the mid-Miocene Randecker Maar appear to be calcareous and marly laminites and limestones of lighter colour (Westphal, 1963; Joachim, 2010; Rasser et al., 2013).

Conclusions

Our study of Diptera larvae and pupae from the Foulden Maar provides new data on the fossil history of aquatic dipterans on the isolated landmass of New Zealand. From this earliest Miocene (~23 ma) lacustrine deposit, we identified several pupal and larval morphotypes of flies, including non-biting midges (of the ingroups Chironominae and Tanypodinae) and phantom-midges (Chaoboridae) as well as several putative pre-imaginal morphotypes of Brachycera. Although widely distributed elsewhere, Chaoboridae have no extant representatives in New Zealand today, indicating that Neogene extinction of some ingroups of Diptera played a role in shaping the extant dipteran fauna in New Zealand. A relative high abundance of aquatic pupae at Foulden Maar is likely the result of taphonomic pathways provided by diatomaceous mats and, perhaps, of the light sediment colour, which facilitated spotting of small fossils such as Diptera pupae in the field. Overall the New Zealand fossil record of merolimnic and other dipterans is still poorly known.

We thank the Gibson family for allowing access to the Foulden Maar site, Nick Butterfield (University of Cambridge) for providing a glass-mounted chironomid larva and Jeffrey Robinson (University of Otago) for providing OU collection numbers. We are grateful to Dagmara Żyła (Leibniz Institute for the Analysis of Biodiversity Change), Hongqu Tang (Jinan University), Galileu Dantas (Instituto Nacional de Pesquisas da Amazônia) and an anonymous reviewer for their efforts in improving our manuscript.

Additional Information and Declarations

Competing Interests

Author Contributions

Data Availability

The authors declare that they have no competing interests.

Viktor O Baranov conceived and designed the experiments, performed the experiments, analyzed the data, prepared figures and/or tables, authored or reviewed drafts of the article, and approved the final draft.

Joachim T Haug analyzed the data, prepared figures and/or tables, authored or reviewed drafts of the article, and approved the final draft.

Uwe Kaulfuss conceived and designed the experiments, performed the experiments, analyzed the data, prepared figures and/or tables, authored or reviewed drafts of the article, and approved the final draft.

The following information was supplied regarding data availability:

All specimens included in this work are housed at the Geology Museum, Department of Geology, University of Otago (OU) with the following accession numbers (OU collection number (original field number)): Tanypodinae: OU46609 (191), OU46626 (208), OU45541 (137), OU44933 (28), OU44930 (23), OU46641 (223); Chironomidae: Chironomini: OU47058 (252), OU46654 (236), OU46645 (227), OU46627 (209), OU46625 (207), OU46620 (202), OU45553 (149), OU45543 (139), OU47051 (245), OU47488 (268); Chironomidae: cf. Chironominae: OU45549 (145), OU46608 (190); Chaoboridae: OU47487 (163), OU46642 (224), OU46631 (213), OU46651 (233), OU46653 (235); Brachycera (?): OU44944 (43), OU44981 (90), OU44982 (91), OU44996 (105), OU45559 (155), OU46644 (226), OU46652 (234), OU46655 (237).

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
