# Peer review of "New records of immature aquatic Diptera from the Foulden Maar Fossil-Lagerstätte, New Zealand, and their biogeographic implications"

_PeerJ, doi:10.7717/peerj.17014_

## Round 0.1 · original submission · Minor Revisions

Please, address all reviewers' comments.

Reviewer 1 ·

Basic reporting

This paper presents very important information and in my opinion should be published. The new data contained in the manuscript are very interesting and original. Results are relevant to the hypothesis. All tables and figures are very suitable for this manuscript, the figures are well done. Literature is sufficient.

Experimental design

The methods used during research are correct and experimental design is well done. One small correction should be made: figures 11 and 12 should be slightly modified – the edge of the marked color field should be more rounded in some places, as can be seen in the photo. The text should be formatted according to PeerJ criteria (for example, see lines 103 and 37).

Validity of the findings

The works presents high scientific value. I propose minor revisions. English should be checked.

·

Basic reporting

The identity of several morphotypes needs to run deeper identification or expand the discussion scope, although indeed hard to do here, e.g., the first taxa, Coelotanypus sp, the author should explain the uniqueness, pupa of Coelopynia also shares those character in the text; Chironominae morphotype 1 can be narrowed down to a genus level. Another aspect, if we use modern morphological characters to infer or calibrate the old fossil, be careful of those ancient groups.

Experimental design

no comment

Validity of the findings

no comment

Additional comments

The MS is worth to be published since the immature Diptera material is extremely rare and hard to obtain. The results provide some fundamental insight into the early fauna formation of Zealandia.

·

Basic reporting

I think the work has scientific merit, with interesting findings. However, there are still some loose ends. For instance, some important references were missing (at least one), there are some inaccuracies in the terminology used in the figures, and, most importantly, one of the key results is problematic in my opinion.
I provided the comments directly on the PDF file.

Experimental design

No comment.

Validity of the findings

One of the results (specifically, the identification of part of the material as Coelotanypus), crucial to support part of the discussion/conclusion is problematic in my opinion; perhaps it is necessary to reevaluate the material and possibly reformulate part of the discussion/conclusions. But, of course, this is just an opinion, and it is up to the authors whether or not to take it into consideration.
I made the comments directly on the PDF file.

---

## Round 0.2 · accepted · Accept

All the reviewers' comments were addressed. The main critique was the taxonomic attribution of the morphotypes. This has been significantly improved and now matches the reviewers' request to expand this section. The language of the manuscript has also been polished according to the reviewer's suggestion. I think that the manuscript is ready for publication.